# Quality Characteristics and Anthocyanin Profiles of Different *Vitis amurensis* Grape Cultivars and Hybrids from Chinese Germplasm

**DOI:** 10.3390/molecules26216696

**Published:** 2021-11-05

**Authors:** Lei Zhu, Xinyue Li, Xixi Hu, Xin Wu, Yunqing Liu, Yiming Yang, Yanqing Zang, Huacheng Tang, Changyuan Wang, Jingyu Xu

**Affiliations:** 1College of Food Science and Technology, Heilongjiang Bayi Agricultural University, Daqing 163319, China; zhulei2580@126.com (L.Z.); lixinyue3676@163.com (X.L.); wuxin12021@163.com (X.W.); liuyunqing0515@163.com (Y.L.); byndzangyanqing@163.com (Y.Z.); byndthc@126.com (H.T.); 2Quality Supervising and Testing Center of Ministry of Agriculture and Rural Affairs for Agricultural Products and Processed Goods, Daqing 163319, China; 3Department of National Coarse Cereals Engineering Research Center, Daqing 163319, China; 4Agri-Food Processing and Engineering Technology Research Center of Heilongjiang Province, Daqing 163319, China; 5Daqing Branch, Heilongjiang Academy of Agricultural Sciences, Daqing 163319, China; huxixi116@163.com; 6Institute of Special Animal and Plant Sciences of Chinese Academy of Agricultural Sciences, Changchun 130112, China; yangyiming@caas.cn; 7College of Agriculture, Heilongjiang Bayi Agricultural University, Daqing 163319, China

**Keywords:** *Vitis amurensis*, anthocyanin composition and content, total phenolic content, general characteristic, antioxidant activity

## Abstract

To evaluate the important *Vitis amurensis* germplasm, the quality characteristics and anthocyanin profiles of the ripe berries of 20 *V. amurensis* grapes and 11 interspecific hybrids in two consecutive years were analysed. Compared with the *V. vinifera* grapes, *V. amurensis* grapes had small berries with low total soluble solids and high titratable acids, and were richer in phenolic compounds except for flanan-3-ols in their skins but had lower phenolic contents in their seeds and showed lower antioxidant activities. An outstanding feature of the *V. amurensis* grapes was their abundant anthocyanin contents, which was 8.18-fold higher than the three wine grapes of *V. vinifera*. The anthocyanin composition of *V. amurensis* was characterized by an extremely high proportion of diglucoside anthocyanins (91.71%) and low acylated anthocyanins (0.04%). Interestingly, a new type of speculated 3,5,7-*O*-triglucoside anthocyanins was first identified and only detected in *V. amurensis* grapes and hybrids. Based on the total phenolic and anthocyanin characteristics, *V. amurensis* grapes were set apart from *V. vinifera* cultivars and the interspecific hybrids, for the same qualities, fell between them, as assessed by principal component analysis.

## 1. Introduction

*Vitis amurensis* Rupr., an important species belonging to the East-Asian population of *Vitaceae*, is mainly distributed in northeast China, Far East Russia and the Korean Peninsula [1]. The cold resistance of *V. amurensis* is the strongest among all the grape species. Its roots and branches can survive at extremely low temperatures of −14–−16 °C and −45 °C, respectively [2]. And *V. amurensis* is highly resistant to white rot (*Coniothyrium diplodiella*), grape anthracnose (*Glomerella cingulata*) and grape bitter rot (*Greeneria uvicola*) [3]. These diseases have very adverse effects on *Vitis vinifera* grapes, which are widely cultivated and commercialized throughout the world. Thus, *V. amurensis* is one of the precious rootstock and breeding resources with resistance to cold and diseases in *Vitis*.

Since the late 1950s, the survey, collection and evaluation of *V. amurensis* germplasm resources have been systematically carried out in China [2]. Through decades of efforts, more than 400 resources are preserved in the *V. amurensis* germplasm repository (VAGR) of Chinese Academy of Agricultural Sciences (CAAS). Moreover, a technical cultivation system of *V. amurensis* cultivars and hybrids has been constructed and promoted in China [4]. In winter, vines do not need burying in most of northern China, which can greatly reduce the cost of cultivation, while the *V. vinifera* grapes can not be cultivated at all in the open fields of Heilongjiang, Jilin and northern Liaoning. Today, China has the largest area in the world cultivated by grapes with the *V. amurensis* pedigree [2]. Most *V. amurensis* resources yield small and red berries [5]. As a result, *V. amurensis* grapes and hybrids are mainly used to make red wines, including dry wines, sweet wines, ice wines and low-alcohol and non-alcoholic wines [5]. The wine pomace can also be used to produce health care products and additives. And a few interspecific hybrids are used as table grapes or juice grapes, in China. However, there are few reports of the quality characteristics of the *V. amurensis* germplasm in international academic journals.

There are rich sources of phenolic compounds in the grapes of *V. amurensis*, including phenolic acids, stibenes, anthocynins, flavonols and flavan-3-ols [6,7,8,9]. In a recent study, a total of 118 phenolic metabolites were detected in wild and cultivated *V. amurensis* grapes collected from Far East Russia [10]. The phenolic extracts from *V. amurensis* have been proven to possess antioxidant [11], anti-inflammatory [12] and antiproliferative [13,14] activities, which could attenuate degenerative processes such as aging [15], cancer [16] and cardiovascular disease [17]. More importantly, phenolic content and composition have significant influences on the quality of grape berries and wines. Among these phenolic compounds, the abundant content of anthocyanins is the most notable characteristic of *V. amurensis* [18]. The literature contains several reports concerning *V. amurensis* anthocyanins from studies using high-performance liquid chromatography coupled with mass detection (HPLC-MS) [8,10,18,19,20,21];however, in these, the selected grape cultivars/accessions were neither comprehensive nor representative.

In this study, the important germplasm resources in VAGR of CAAS, including 20 *V. amurensis* cultivars/accessions and 11 interspecific hybrids crossed with *V. vinifera*, were collected to systematically and comprehensively investigate their quality characteristics and, especially, their anthocyanin profiles, and comparedwith 3 red wine grapes of *V. vinifera.* Analysis by ultra-performance liquid chromatography coupled with Q-exactive orbitrap mass detection (UPLC/Q-Exactive orbitrap MS) was conductedon their anthocyanin profiles, which are more efficient and sensitive techniques in comparison with HPLC-ion trap MS. Rich anthocyanin compounds were found and a new type of anthocyanin was preliminarily identified in grapes of the *V. amurensis* pedigree.

## 2. Material and Methods

### 2.1. Plant Material

All the materials of 31 *V. amurensis* cultivars/accessions and hybrids in this study were collected from the VAGR of CAAS. The materials of the three *V. vinifera* wine grapes were collected from the experimental vineyards of China Agricultural University (Table 1). All red berry samples were harvested upon ripening in two consecutive years, 2017 and 2018, determined based on their content of total soluble solids and as judged from seeds’ color-change to dark brown without the senescence of berry tissue. At least 30 clusters were collected at 6 grapevines for each cultivar. The fresh clusters were surrounded by ice packs and immediately taken to the laboratory. Then, three 100-berry batches, with berries randomly selected from the top, middle and bottom portions of the clusters, were collected. Each batch was considered to be one sample, too allow for three experimental replications for each grape cultivar/accession.

### 2.2. Chemicals and Standards

Folin and Ciocalteu’s phenol reagent (2 *N*), 2,2-diphenyl-1-picrylhy-drazyl (DPPH) (≥97%), 2,2′-azino-bis-(3-ethylben zothiazoline-6-sulfonic acid) diammonium salt (ABTS) (≥98%), 2,4,6-tripyridyl-s-triazine (TPTZ) (≥99%) and 6-Hydroxy-2,5,7,8-tetramethylchro-man-2-carboxylic acid (Trolox) (≥98%) were obtained from Aladdin (Shanghai, China). The standards, gallic acid (99%), catechin (≥95%) and rutin (98%), used in the colorimetric determination were purchased from Macklin (Shanghai, China), and the anthocyanin standards for UPLC-MS, malvidin-3-*O*-glucoside (≥90%) and malvidin-3,5-*O*-diglucosides (≥90%) were obtained from Sigma-Aldrich (St. Louis, MO, USA). HPLC grade reagents, formic acid and acetonitrile were purchased from Fisher Scientific Co. (Fairlawn, NJ, USA). All other analytical-grade chemicals were purchased from Chinese Reagent Network (http://www.labgogo.com/, accessed on: 1 November 2018).

### 2.3. Determination of General Berry Characteristics

Fifty fresh berries from each replicate sample were randomly selected to determine their general characteristics. The weights, transverse diameters and longitudinal diameters of the selected berries were recorded (*n* = 150). Approximately 20 g berries of eachbatch’s 50 remaining berries were then weighed and crushed to assay their titratable acids (*n* = 3). The crushed volumes were diluted to 200 mL with distilled water. After centrifugation, 20 mL of supernatant was titrated with 0.1 mol/L NaOH to a pH endpoint of 8.1 [22], and the residue berries were crushed to directly determine the total soluble solid (*n* = 3) with a hand-held refractometer (Wancheng Co., Beijing, China) and the pH (*n* = 3) with a pH meter (S210-B, METTLER TOLEDO, Zurich, Switzerland).

### 2.4. Pretreatment and Extraction of Phenolic Compounds

The skins and seeds were manually separated, freeze-dried (Alpha 1–4LSCbasic, Martin Christ Co., Osterode, Germany) and ground (BLF-YB2000, Bailifu Co., Shengzhen, China) into a fine powder. The ground seeds were defatted, twice, with petroleum ether for 3 h. The final skin and seed samples were stored in vacuum-packaged polyethylene pouches at −80 °C for subsequent analysis.

A protocol developed in the previous work [23] was used for the extraction of phenolic compounds. Briefly, the solvents of methanol/water/acetic acid (70:29:1, *v*/*v*/*v*) was used for the skins, and methanol with 0.1% acetic acid for the seeds. The pulverized powder (0.5 g), mixed with 20 mL of extraction solvent, was shaking-extracted in a shaker (SHZ-88A, Taicang Experiment Equipment Factory, Taicang City, China) at room temperature for 2 h. After centrifugation (GL-21M, Xiangyi Co., Hunan, China), the residues were re-extracted twice and thee extracts from each sample were collected together. The skin extracts for each replicate sample, after removing a volume of 2 mL for use in the colorimetric determination, was rotatory evaporated (RE-52A, Yarong Biochemistry Instrument Factory, Shanghai, China) at 30 °C and redissolved in the extraction solvent to a unified volume of 5 mL for the anthocyanin analysis.

### 2.5. Determination of Total Contents of Phenolic Compounds

The total phenol content (TP) was assayed with the Folin–Ciocalteu colorimetric method [24] with a UNICO UV-2100 spectrometer (UNICO, New York, NY, USA). The absorbances were converted to TP expressed as mg gallic acid equivalents (GAE)/g of dry weight (DM). The total flavonoid content (TFO) was measured with the NaNO_2_-AlCl_3_ method [25]. The TFO was expressed as mg rutin equivalents (RAE)/g DM, and total flavan-3-ols content (TFA) was determined by the Vanillin method [26] and expressed as mg catechin equivalents (CAE)/g DM.

### 2.6. Analysis of Anthocyanin Compounds

The anthocyanins in grape skins were analyzed using an Ultimate 3000 UPLC/Q-Exactive orbitrap MS (Thermo Fisher Scientific, Waltham, MA, USA). The samples were diluted 10 times with an aqueous solution containing 10% acetonitrile and 2% formic acid and injected (5 µL), directly after filtration and through a 0.45-µm inorganic membrane, onto a Thermo GOLD HYPERSIL column (C18, 50 mm × 2.1 mm, 1.9 µm) at 22 °C. The solvent system consisted of an aqueous solution containing 2% formic acid (phase A) and an acetonitrile solution containing 2% formic acid (phase B), and the flow rate was 0.3 mL/min. The gradient profile was from 2% to 10% B for 3 min, from 10% to 20% B for 3 min, isocratic 20% B for 1 min, from 20% to 100% for 5 min and isocratic 100% for 2 min. MS analysis was conducted using heating electrospray ionisation (HESI), positive ion model, a 13-psi nebulizer pressure, 35-mL/min dry gas flow rate, 300 °C dry gas temperature, and 160–1000 *m*/*z* scan range. Monoglucoside and diglucoside anthocynins were quantified using malvidin-3-*O*-glucoside and malvidin-3,5-*O*-diglucoside as standards, respectively, and expressed as μg MGE or mg MGE per g DM of grape skins.

### 2.7. Determination of Antioxidant Activities

The antioxidant activities of the phenolic extracts from grapes’ skins and seeds were based on DPPH free-radical-scavenging ability [27], ABTS free-radical-scavenging ability [28] and ferric reducing/antioxidant power (FRAP) [29]. The results were expressed as µmol Trolox equivalents [30].

### 2.8. Statistical Analysis

All results were expressed as means ± standard deviations (SD), as all the data of the variances were compatible with the normal distribution, but most were not homogeneous. The data from the same indicator between grape groups were subjected to Welch’s analysis of variance (ANOVA) with the Games–Howell test at a 95% confidence level. The correlations of different indicators were analyzed with Pearson correlation coefficient at a 95% confidence level. The principal component analysis (PCA) was performed to investigate the phenolic-based relationships of different grape cultivars/accessions. All the statistical analyses were conducted with SPSS 19.0 (SPSS Inc., Chicago, IL, USA).

## 3. Results and Discussion

### 3.1. General Characteristics

In general, the *V. amurensis* resources assayed in this study, including wild accessions/cultivars and intraspecific hybrids, had lighter and smaller berries with low total soluble solids and higher titratable acids. The three *V. vinifera* grapes with heavier and bigger berries produced higher total soluble solids and lower titratable acids. As expected, the interspecific hybrids were intermediate between *V. amurensis* and *V. vinifera* (Figure 1, Appendix A); but there were also several exceptions. Among all the assayed grapes, the interspecific hybrid ‘Huapu-1’ had the weightiest and biggest berries. The intraspecific hybrids of *V. amurensis,* ‘Shuanghong’ and ‘Shuangfeng’, had higher total soluble solids and lower titratable acids than some interspecific hybrids.

### 3.2. Total Contents of Phenolic Compounds in Grape Skins and Seeds

The average TP in grape skins of the three grape groups varied, and were, in decreasing order, Am > Vi > Hy (Figure 2a, Appendix A). The TP in the skins of the grape cultivars/accessions of *V. amurensis* were generally higher. The accession ‘85010’ had the highest TP (60.24 mg GAE/g DM) among all the grapes collected, followed by ‘14’, ‘92’, ‘Tonghua-1’ and ‘086919’,while the lowest TP was found in the interspecific hybrid ‘Huapu-1’ (16.45 mg GAE/g DM). Also, many interspecific hybrids possessed lower TP, namely ‘Beihong’, ‘Xuelanhong’, ‘Gongniang-1’ and ‘Zuohong-1’. The TP in skins also varied significantly within the same group. For example, the *V. amurensis* accession ‘75081’ had lower TP, which was similar with the interspecific hybrid ‘Zuohong-1’. Similar distributions were found in the skin TFOs among all the grapes studied (Figure 2b, Appendix A). However, the TFA in the skins of *V. amurensis* grapes and hybrids were significantly lower than in the three *V. vinifera* grape skins (Figure 2c, Appendix A).

The total content of phenolic compounds, including TP, TFO and TFA, in the seeds presented with consistent variation, by decreasing order of Vi > Hy> Am (Figure 2a–c, Appendix A). However, there were also significant differences among grape seeds within the same group. Taking the *V. amurensis* for example, the TP, TFO and TFA in the accessions ’84,001’ and ’84,002’ were significantly higher than those in other grapes, and were two-fold more than accessions ’86,919’ and ‘92’, which had the least total phenolic contents.

### 3.3. Anthocyanin Profiles

#### 3.3.1. Identification of Anthocyanin Compounds

A UPLC/Q-Exactive orbitrap MS was applied to the qualitative and quantitative analysis of anthocyanin compounds in this study. Compared with HPLC, UPLC can separate mixtures more efficiently so as to dramatically shorten the elution time in the chromatographic column, and the Q-Exactive orbitrap MS can detect more micro-components in samples. Combined with literature data [18,31,32], the compounds were identified by means of the extracted ion chromatograms obtained in MS^2^ mode.

A total of 48 anthocyanins were detected across all studied samples (Table 2). All the anthocyanins were glucoside derivatives of six anthocyanidins: delphinidin (Dp), cyanidin (Cy), petunidin (Pt), pelargonidin (Pg), peonidin (Pn) and malvidin (Mv), which were dependent on the observed *m*/*z* characteristic fragmentation values of 303, 287, 317, 271, 301 and 331, respectively. The simple glucoside derivatives, monoglucoside and diglucoside anthocyanins, were identified on the basis of their molecular ions [M]^+^ as well as the fragment ions corresponding to the anthocyanidin after cleavage of one glucose unit [M-162]^+^ and two glucose units [M-162]^+^ and [M-324]^+^ in MS^2^ mode, respectively. While the acylated derivatives, acetyl, *p*-coumaroyl and caffeoyl anthocyanins were confirmed by their molecular ions [M]^+^ as well as the fragment ions corresponding to the anthocyanidin after cleavage of one esterified glucose moiety of [M-204]^+^, [M-308]^+^ and [M-324]^+^, in MS^2^ mode and respectively.

A new class of anthocyanins, eluted at the earliest (2–4 min, Table 2), was detected in all the *V. amurensis* grapes and hybrids, but not in the three *V. vinifera* grapes. These molecular ions [M]^+^ were 789, 773, 803, 787 and 817 in chronological order, and their mass spectrum were characterized by the three fragment ions of [M-162]^+^, [M-324]^+^ and [M-486]^+^ caused by the stepwise cleavage of three glucose units (Figure 3a). In the chemical structures of anthocyanidins there are three phenolic hydroxyl groups at the positions C3, C5 and C7, respectively. In grapes, the monoglucoside anthocyanin (3-*O*-glucoside) is produced by the glucosylation of one hydroxyl group at C3, while the diglucosides anthocyanin (3,5-*O*-diglucosides) is produced by the glucosylation of the two hydroxyl groups at C3 and C5, respectively. Although 3,7-diglucosides of anthocyanidins, shown to exist in some flowers [33], have been detected in red wine, this was not entirely certain [34]. These compounds should not be the caffeoyl diglucosides anthocyanins, which possess similar molecular ions and are eluted behind all the simple glucoside anthocyanins and acetyl diglucosides anthocyanins. So, we preliminarily speculate that these new anthocyanins were the 3,5,7-*O*-triglucosides of Dp, Cy, Pt, Pn and Mv, respectively. In addition, a *p*-coumaroyl triglucoside anthocyanin ([M]^+^, 963) was identified in all the interspecific hybrids except for ‘Zuohong-1’ and ‘Zuoyouhong’, and was assigned to Mv-3-*O*-(6-*O*-*p*-coumaryl)-5-*O*-7-*O*-triglucosides by the loss of two glucosides (*m*/*z* 801, 639) and a coumaroyl glucoside (*m*/*z* 331) (Figure 3b). To our knowledge, this is the first time the detection of triglucoside anthocyanins in grapes has been reported. However, further research is still needed to confirm this speculation.

Under the condition of separation and detection in this study, two couples of anthocyanins with the same molecular ions [M]^+^ were eluted almost at the same time. In other words, multiple fragment ions contributed to the final intensity of a molecular ion signal in the mass spectrum. For example, Dp-3-*O*-(6-*O*-*p*-coumaryl)-5-*O*-diglucoside ([M]^+^, 773) was indistinguishable from Cy-3-*O*-(6-*O*-caffeoyl)-5-*O*-diglucoside (Table 2, Figure 3c). Similarly, Pt-3-*O*-(6-*O*-*p*-coumaryl)-5-*O*-diglucoside ([M]^+^, 787) was isobaric with Pn-3-*O*-(6-*O*-caffeoyl)-5-*O*-diglucoside (Table 2, Figure 3d). However, the fragment ion abundance of *p*-coumaroylation was obviously stronger than that of caffeoylation, which indicated that caffeoyl derivatives were few and insignificant.

#### 3.3.2. Total Anthocyanin Contents

The total anthocyanin contents (TA) varied significantly among different grape groups (Figure 4a, Appendix A). The *V. amurensis* grapes had the highest average TA (31.83 mg MGE/g DM). The *V. amurensis* accession ‘086919’ had the most TA (45.13 mg MGE/g DM) among all the grapes assayed, followed by ’85,010’, ‘92’, ‘14’, ‘Changbai-5’, ‘Zuoshan-1’ and ‘Zuoshan-2’, with more than 40 mg MGE/g DM. The lowest average TA (3.60 mg MGE/g DM) was found in the three *V. vinifera* cultivars, and the hybrids’ contents (10.07 mg MGE/g DM) fell between *V. amurensis* and *V. vinifera*.

#### 3.3.3. Composition of Anthocyanins

Among the derivatives of different anthocyanidins, Mv-derivatives were the most abundant components in the three grape groups, which was in agreement with our previous research on *V. vinifera* cultivars [35] and East Asian species [18].The average percentages of Mv-derivatives in the skins of *V. amurensis* grapes (58%) and hybrids (56%) were significantly less than that in the three *V. vinifera* grape skins (80%) (Figure 4b), and was approximate for the average percentages of Pn-derivatives among the *V. vinifera* grapes (10%), *V. amurensis* grapes (8%) and hybrids (10%), while the average percentages of Dp-derivatives, Cy-derivatives and Pt-derivatives in *V. amurensis* grapes (15%, 9% and 9%) and hybrids (19%, 6% and 9%) were obviously higher than those in the three *V. vinifera* grapes (4%, 1% and 5%). In addition, it has been confirmed that Pg-derivatives are present at trace concentrations in some grape cultivars of *V. vinifera* [36,37], *V. amurensis* [10,21], *V. labrusca*, *V. aestivalis* [18] and *V. rotundifolia* [38]. In this study, pg-derivatives were found in all the grapes assayed, accounting for 0.00046–0.10% of TA (Figure 4b). Only two simple glucoside derivatives, Pg-3-*O*-glucoside and Pg-3,5-*O*-diglucosides, were detected (Table 2). However, there was a significant difference in the anthocyanidin compositions among the different grapes. For example, the interspecific hybrids ‘Hasang’ and ‘Zuohong-1’, as well as the *V. amurensis* accession ‘Tonghua-1’, had significantly smaller proportions of Mv-derivatives (2%, 26% and 12%) than the other grapes, while ‘Hasang’ had the highest percentages of Dp-derivatives (83%) among all the grapes assayed, followed by ‘Tonghua-1’ (55%) and ‘Zuohong-1’ (45%).

The six kinds of anthocyanidin derivatives can be divided according to the positions and numbers of the hydroxyl and methoxyl substituents in their B rings [39]. The degree of anthocyanins’ hydroxylation affects the hue and stability of their colors. The more hydroxyl groups in their B rings, the more blue anthocyanins appear [39]. In this study, the trihydroxylated anthocyanins were the richest in all the grape skins collected. There was no difference the average percentages of trihydroxylated anthocyanins among the *V. vinifera* grapes (88%), *V. amurensis* grapes (83%) and interspecific hybrids (84%) (Figure 4c). More methoxyl groups in anthocyanins’ B rings contribute more redness [40], and methylated derivatives (Pt-, Pn- and Mv-derivatives) were dominant in most grapes (Figure 4d) except in ‘Hasang’, ‘Zuohong-1’ and ‘Tonghua-1’. The average percentages of methylated derivatives were lower in *V. amurensis* grapes (76%) and hybrids (75%) compared with the three *V. vinifera* grapes (95%) (Figure 4c).

There was a distinct separation in the glucosylated composition of the anthocyanins of the different grape groups (Figure 4e). The diglucoside anthocyanins were predominant in the *V. amurensis* grapes (91.71% of TA on average), which was consistent with the results of previous studies [18,19,21]. The average percentage of diglucoside anthocyanins was significantly less in the hybrids of *V. amurensis* and *V. vinifera* (63%), but there was an obvious difference in glucosylated composition among the interspecific hybrids. The percentage of diglucosides anthocyanins ranged from 33%, in ‘Zuohong-1’, to 86%, in ‘RS’. In the three *V. vinifera* wine grapes, almost all of anthocyanins were monoglucoside derivatives. In addition, the speculated triglucoside anthocyanins found in *V. amurensis* grapes and hybrids were present only in trace amounts.

Razgonova et al. [10] identified 7 acylated derivatives, out of a total of 18 anthocyanin compounds, in *V. amurensis* grapes from Far East Russia. In this study, a richer group of acylated anthocyanin compounds (31) were detected (Table 2). However, it has been reported that *V. amurensis* grapes snone [18,21] or a few [41] acylated anthocyanins. A similar result, of only 0.40% content acylated anthocyanins of TA in all the *V. amurensis* grapes on average (Figure 4f), was obtained in this study. In *V. amurensis* grapes, acetyl anthocyanins (0.38%) were the main and common acylated type; only trace coumaryl anthocyanins were found. In the interspecific hybrids, acetyl, coumaryl and caffeoyl anthocyanins accounted for 2%, 5% and 0.05% of TA on average and respectively, but acylation varied widely, due to the samples’ different pedigrees. The contents and compositions of acylated anthocyanins in ‘Zuohong-1’ and ‘Zuoyouhong’ were extremely similar with *V. amurensis* grapes. An outstanding exception was ‘Hasang’, with the highest percentage of acylated anthocyanins (51%) among all the assayed grapes, while ‘Beichun’ (9%), ‘Huapu-1’ (9%) and ‘RS’ (7%) also had relatively higher percentage of acylated anthocyanins. ‘Beichun’ had the highest percentage of acetyl anthocyanins(7%) among all the hybrids, while coumaryl were significantly more than acylated anthocyanins in ‘Hasang’, ‘Huapu-1’, ‘RS’ and ‘Gongniang-1’. In the three *V. vinifera* grapes, the acylated anthocyanins accounted for higher proportions (16%) than those in *V. amurensis* grapes and hybrids on average, which was similar to the results of Liang et al. [35,41].

### 3.4. Antioxidant Properties of Phenolic Extracts from Grape Skins and Seeds

In this study, the antioxidant capacities (Figure 2d–f, Appendix A), in vitro, were assayed by DPPH, ABTS and FRAP. For the seeds, the variation of the average antioxidant values among grape groups showed a similar tendency with respect to total contents of phenolics. The average values of antioxidant indicators were significantly lower in the *V. amurensis* grapes and hybrids than in the *V. vinifera* grapes. For the skins, the average level of antioxidant activities decreased in the order of Vi > Am > Hy, which differed from the distribution of total phenolic contents; but the difference of antioxidant values was not significant between *V. vinifera* and *V. amurensis*.

### 3.5. Correlation Analysis

The total phenolic contents in the seeds presented a more obvious consistency as compared with the skins, which maybe because the seed phenolics mainly consisted of flavan-3-ols [30], while the skin phenolics contained more variety, such as phenolic acids, flavonols, anthocyanins and flavan-3-ols [18]. This was confirmed by the correlation analysis among the total phenolic contents. There were extremely high correlations in the seeds (0.997–0.999, Appendix A), while in the skins, the correlations were positive and significant among TP, TFO and TA (0.644–0.963) and TFA had low correlations with TP (0.036) and TFO (0.060) and a negative correlation with TA (−0.500, Appendix A). It was indicated that the main phenolic type was anthocyanins rather than flavan-3-ols (mainly tannins) in the skins of red wine grapes collected in this study, especially *V. amurensis*.

Correlation analysis was also performed between samples’ phenolic contents and antioxidant activities. Generally, the richer the phenolic content grape materials possess, the stronger the antioxidant activities their phenolic extracts [30,42,43]. The correlations in the grape seeds were extremely significant (0.965–0.989, Appendix A), and there were also significant correlations (0.257–0.896, Appendix A) except for the coefficients of TFA and FRAP (0.197). The differences between the seeds and skins should be due to the complex composition of phenolic compounds in skins and the low flavan-3-ols in *V. amurensis* grape skins. Among the anthocyanin compounds, only the simple diglucoside derivatives, the most abundant type in *V. amurensis* grapes, significantly contributed to the antioxidant capacities of the skin phenolic extracts (0.237–0.471, Appendix A).

### 3.6. Principle Component Analysis

In order to separate these grape cultivars/accessions and describe the phenolic characteristics of different grape groups, a principal component analysis (PCA) was carried out on the basis of anthocyanin composition, total contents and the antioxidant activities of their phenolic compounds. A total of six principal components with eigenvalues >1 were obtained, explaining 91.69% of total variance. PC1 and PC2 possessed relatively high percentages of variance (36.32% and 20.06%, respectively). PC1 was mainly represented by total contents and antioxidant activities of seed phenolic compounds, skin TFA and the percentage of acetylated anthocyanins with positive correlations, as well as TA, TP, TFO and the percentages of diglucosides anthocyanins and Cy-derivativesin skins with negative correlations. PC2 consisted primarily of the antioxidant activities of the skins’ phenolics and the percentage of Dp-derivatives, coumaryl and caffeoyl anthocyanins with positive PC2 values, as well as nonacylated and methylated anthocyanins with negative PC2 values (Figure 5a).

A scatter plot on the basis of PC1 and PC2 (Figure 5b) showed that the cultivars/accessions from the same group appeared to occupy a certain PC1 range. The *V. amurensis* cultivars/accessions, located on the negative side of the PC1 axis, were characterized by having high TA and proportions of diglucoside anthocyanins. The three *V. vinifera* grapes were located on the far positive side of the PC1 axis, mainly because of their high total contents and the antioxidant capacities of the phenolic compounds in their seeds, as well as the high TFA and proportions of acetylated anthocyanins in their skins. As expected, the hybrid grapes were located between the *V. amurensis* and *V. vinifera* grapes on the PC1 axis, with a value range of −0.53–1.20. There was an exception—the hybrids ‘Hasang’ had an extremely high PC2 values (4.25) compared with other grapes (−1.41–1.53), due to its high proportions of Dp-derivatives, coumaryl and caffeoyl anthocyanins.

## 4. Discussion

### 4.1. Selection of Core V. amurensis Germplasm

There are abundant *V. amurensis* germplasm resources in China. Through domestication and selection of wild resources, a number of *V. amurensis* cultivars have been screened out. In 1957, the female flower lines of ‘Changbai’ and ‘Tonghua’ were selected, outstandingly represented by ‘Changbai-9’ and ‘Tonghua-3’. It was a very exciting discovery that ‘Changbai-11’ was a hermaphrodite flower cultivar, and officially named as ‘Shuangqing’ in 1975. It was the first ever finding of a *V. amurensis* resource with hermaphrodite flowers, which ended the commercial production of female flower cultivars with the male pollenizers [44]. Other female flower cultivars, ‘Zuoshan-1’ and ‘Zuoshan-2’, were successfully selected in the middle of the 1980s. Based on the wild *V. amurensis*, a series of intraspecific hybrids with bisexual flowers were bred from the female parents of pistillate flower genotypes and a male parent of ‘Shuangqing’, such as ‘Shuangfeng’ [45], ‘Shuangyou’ [46] and ‘Shuanghong’ [47]. Through the crossing between *V. amurensis* and *V. vinifera*, plenty of new interspecific hybrids were selected from the filial generations, including the red wine cultivars ‘Zuohong-1’ [48], ‘Zuoyouhong’ [49], ‘Xuelanhong’ [50], ‘Beichun’ [51], ‘Beihong’, ‘Beimei’ [52], ‘Gongniang-1’, ‘Gongniang-2’ [53] and ‘Huapu-1’ [54], as well as the ice wine cultivar ‘Beibinghong’ [55]. Compared with *V. amurensis* grapes, the interspecific hybrids had better qualities for making wine, such as relatively high sugar and low acid [5]. These hybrids considerably enhanced production and promoted the industry’s development of *V. amurensis* [5]. In this study, the above representative *V. amurensis* cultivars and hybrids and important accessions from different regions were collected from the VAGR of CAAS to investigate the quality characteristics and anthocyanin profiles of the overall *V. amurensis* germplasm in China.

### 4.2. General Characteristics and Total Phenolic Contents of V. amurensis Grapes and Hybrids

The mature berries of *V. amurensis* collected in this study were round and small, which matched the descriptions of Liu and Li [5] and Chen et al. [6]. Among the 20 studied *V. amurensis* grapes, their transverse diameters were only 0.03 mm (Appendix A) smaller than their longitudinal diameters, on average. However, the obtained range of 9.84–13.15 mm of the diameters in this study was longer than that of 5–12 mm, measured by Zhuang [56]. Our average berry weights (1.02 g, Appendix A) were heavier than the 10 *V. amurensis* accessions (0.83 g) studied by Liang et al. [41]. *V. amurensis* grapes have beenproven to produce high acid and relatively low sugar [5]. In this study, the total soluble solids were 9.56 Brix to 17.22 Brix, which was similar to the sugar content range (80–170 g/L) concluded by Jie [57]. The observed titratable acids ranged from 0.67% to 2.04%, which were lower than the organic acid contents (15–30 g/L) concluded by Jie [57], mainly because of the relatively lower acid content of six *V. amurensis* cultivars, ‘Zuoshan-1’, ‘Zuoshan-2’, ‘Shuangqing’, ‘Shuangyou’, ‘Shuanghong’ and ‘Shuangfeng’, that fall a range of 0.67–1.27%. The differences in the general characteristics were perhaps induced by the various genotypes and terroir.

The distribution of phenolic contents between seeds and skins were different between *V. amurensis* and *V. vinifera*. The TP and TFO contents in skins were higher than in seeds for most *V. amurensis* cultivars/accessions, with a few exceptions; the *V. vinifera* grapes had much higher total phenolic contents in their seeds than in their skins, which agrees with that found in other studies [42,58,59]. The performances among interspecific hybrids were different due to various breeding parents, butfor all the grapes assayed, TFA was much richer in seeds than in skins. In general, the *V. amurensis* grapes collected in this study had richer phenolic compounds in their skins, except for flavan-3-ols, but lower phenolic contents in their seeds compared with *V. vinifera* grapes.

### 4.3. Anthocyanin Profiles of V. amurensis Grapes and Hybrids

There is abundant anthocyanin content in *V. amurensis*. In this study, the average TA in *V. amurensis* grape skins was 8.18-fold higher than that in the skins of the three *V. vinifera* wine grapes. In previous studies, the levels of TA in entire *V. amurensis* samples was comparatively high compared with *V. vinifera* [35,41], other East Asian species [18] and American species [19,41].

A proportion of more than 90% diglucoside anthocyanins is the most important trait in the anthocyanin composition of *V. amurensis*. Diglucoside anthocyanins are more stable, compared with monoglucoside anthocyanins, due to their molecular structures, but diglucoside anthocyanins in wines form stable and complex pigments with relative difficulty, compared with monoglucoside anthocyanins, such that aged wines with a certain amount of diglucoside anthocyanins are more susceptible to browning and retain less color [39,60]. In addition, a significantly smaller proportion of acylated anthocyanins was found in *V. amurensis* compared with *V. vinifera.* Acylation may increase the stability and solubility of anthocyanin molecules [39]. As a result, *V. amurensis* grapes and hybrids are poorly suited to the production of aged red wines. Together with high acids and low sugar, *V. amurensis* grapes and hybrids are mainly suitable for sweet wines. Thus, grape breeders in China have been devoted to improving the wine-making quality of *V. amurensis* by the crossbreeding it with *V. vinifera*.

Grapes possess a rich variety of anthocyanin compounds because of the functions of modification enzymes in anthocyanin biosynthesis, such as hydroxylases, methyltransferases, glucosyltransferases and acyltransferase. The compounds 5-*O*-glucosyltransferases (5GT) are responsible for converting monoglucoside anthocyanins into diglucoside anthocyanins. It was reported for the first time that the nonfunctional 5GT allele in the No. 9 chromosome (chr 9) might result in the inability of *V. vinifera* grapes to synthesize diglucoside anthocyanins [61]. Through the cloning and analysis of the *5GT* allele in chr 9 from *V. vinifera* and other 7 *Vitis* spcies, Yang et al. [43] found that 27 functional 5GT alleles were present in non-*V. vinifera Vitis* species, and 18 out of 26 5GT alleles were apparently nonfunctional in *V. vinifera*. He et al. [62] isolated successfully a *5GT* allele in chr 9 from *V. amurensis* ‘Zuoshan-1’, whose recombinant proteins were confirmed to have a *5GT* function by enzymology tests; and a functional *5GT* allele was found in chr 5 of *V. vinifera* ‘Cabernet Sauvignon’ [63], which might explain the trace diglucosides anthocyanins (0.23% on everage) detected in the three *V. vinifera* grapes of this study. However, the biosynthesis and regulation mechanisms of diglucoside anthocyanins need to be explored father, as it could provide a theoretical basis for the reduced proportion of diglucoside anthocyanins in *V. amurensis.*

It is interesting that a new class of anthocyanins was found in all the examined *V. amurensis* grapes and hybrids. According to its mass spectra, it is preliminarily speculated that these anthocyanins are the 3,5,7-*O*-triglucosides of five anthocyanidins (Dp, Cy, Pt, Pn and Mv). Also, a *p*-coumaroyl triglucosides of Mv was detected in most of the interspecific hybrids. To our knowledge, this is the first time the detection of triglucosides anthocyanins in grapes has been reported, and the above-mentioned new anthocyanin type has not been found in other plant species. The speculated triglucoside anthocyanins might be the analytic targets for identifying grapes and wines with the *V. amurensis* pedigree, but further research is still needed to confirm this speculation.

## 5. Conclusions

In this study, we investigated and analyzed the berry qualities of important and representative *V. amurensis* grapes and interspecific hybrids maintained in the VAGR of CAAS. Three red wine grapes of *V. vinifera,* widely cultivated throughout the world, were also collected as a comparative group. The survey items included general characteristics, total contents and antioxidant activities of phenolic compounds, and especially of anthocyanin contents and compositions. In general, the anthocyanins of *V. amurensis* grapes featured rich total contents, high diglucosylation and low acylation levels. With the application of UPLC/Q-Exactive orbitrap MS, a total of 48 anthocyanins were detected and a new type of speculated 3,5,7-*O*-triglucoside anthocyanins were identified in grapes with the *V. amurensis* pedigree. In addition, the ripe *V. amurensis* berries had higher contents of titratable acids, total phenols and flavonoids in skins, and lower total soluble solids, phenolics in seeds, flanan-3-ols and antioxidant activities. Through correlation analysis, the contents of total phenolic compounds (except for flanan-3-ols in skins) and simple diglucosides anthocyanins obviously contributed to the antioxidant activities of phenolic extracts. Most characteristics of the interspecific hybrids fell between those of *V. amurensis* and *V. vinifera*, which indicated interspecific breeding with *V. vinifera* could improve the wine-making quality of *V. amurensis*. This work is an important component in our ongoing effort to develop a comprehensive database the qualities and nutritional characteristics of *V. amurensis* germplasms for breeding and commercialization.

## Figures and Tables

**Figure 1 molecules-26-06696-f001:**
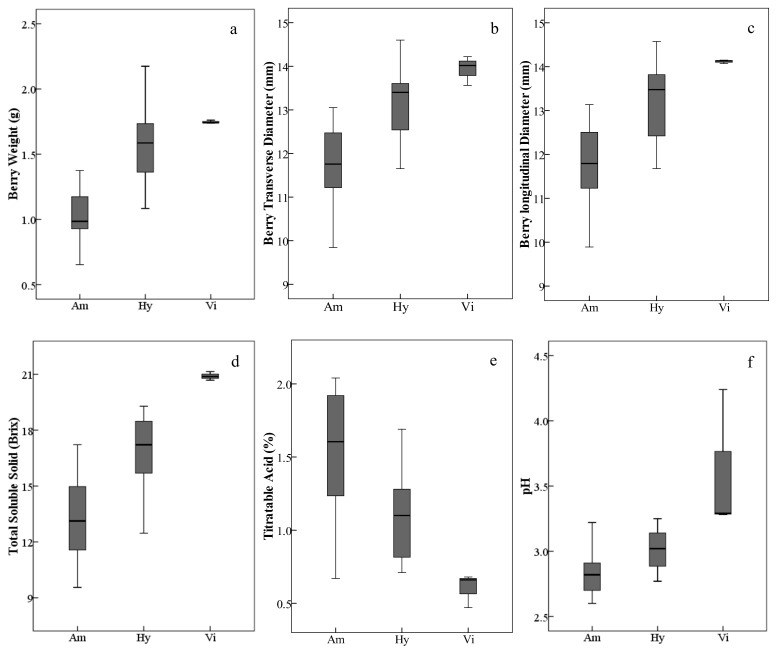
Ranges and distributions of weight (**a**), transverse diameter (**b**), longitudinal diameter (**c**), total soluble solid (**d**), titratable acid (**e**) and pH (**f**) of ripe berries belonging to different grape species/groups harvested in 2017 and 2018. The height of each box is equal to the interquartile range. The horizontal line in the interior of each box is the median. The whiskers extended from the top and bottom of each box with horizontal lines are the maximums and minimums, respectively. Abbreviations: Am, *V. amurensis*; Hy, hybrids of *V. amurensis* and *V. vinifera*; Vi, *V. vinifera*.

**Figure 2 molecules-26-06696-f002:**
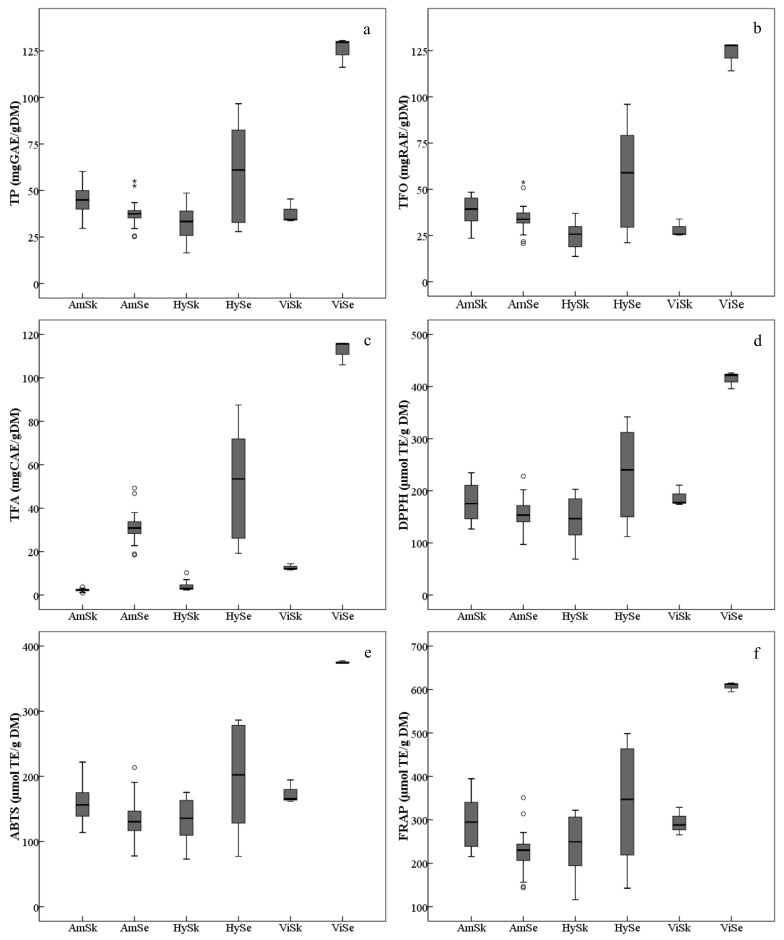
Ranges and distributions of total phenolics (TP, (**a**)), total flavonoids (TFO, (**b**)), total flavan-3-ols (TFA, (**c**)), DPPH free radical scavenging activities (DPPH, (**d**)), ABTS free radical scavenging activities (ABTS, (**e**)) and ferric ion/reducing antioxidant powers (FRAP, (**f**)) of ripe berry skins (Sk) and seeds (Se) belonging to the three grape groups from 2017 and 2018. The hollow dots and asterisks outside the whiskers indicate the mild and extreme outliers, respectively. The abbreviations of grape groups follow Figure 1.

**Figure 3 molecules-26-06696-f003:**
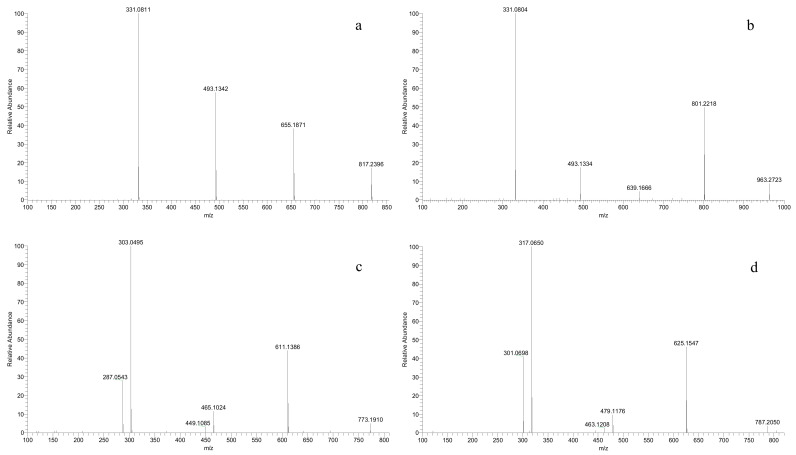
MS^2^ spectrum of the ion signals *m*/*z* 817 (**a**), 963 (**b**), 773 (**c**) and 787 (**d**).

**Figure 4 molecules-26-06696-f004:**
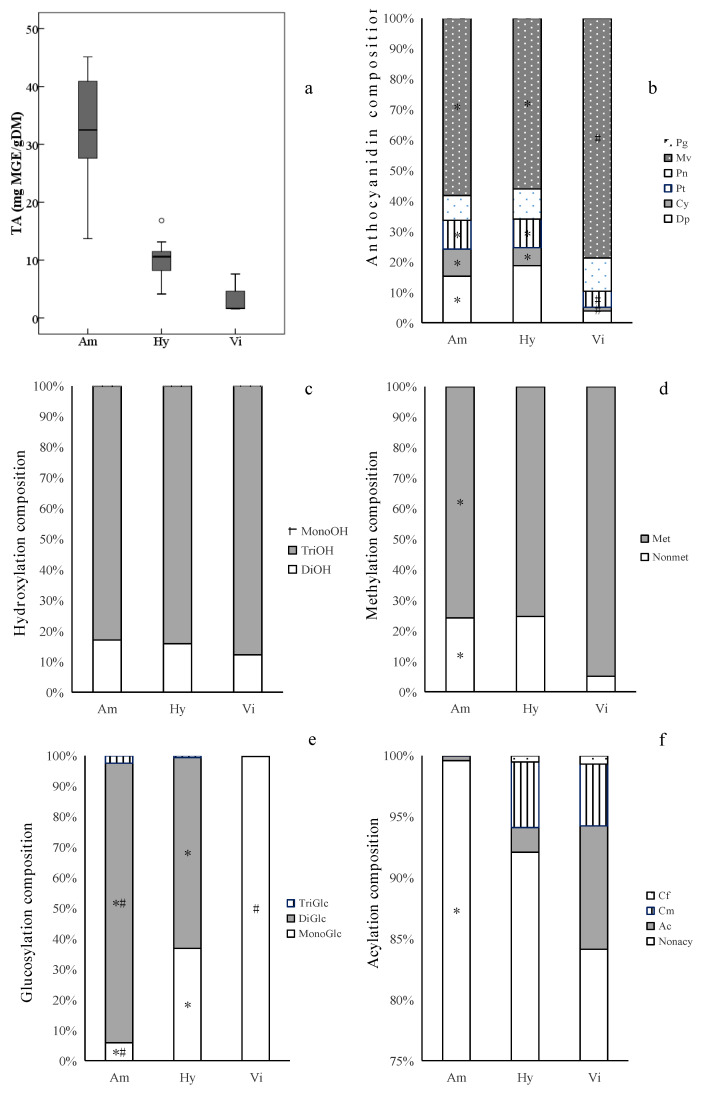
Ranges and distributions of total anthocyanins (TA, **a**) and anthocyanin compositionsof anthocyanidin (**b**), hydroxylation (**c**), methylation (**d**), glucosylation (**e**) and acylation (**f**) in ripe berry skins belonging to the three grape groups from 2017 2018. * indicates significant difference (*p* < 0.05) vs. *V. vinifera* grapes for each indicator by Welch’s ANOVA with the Games–Howell test. ^#^ indicates significant difference (*p* < 0.05) vs. interspecific hybrids. Abbreviations: Dp, delphinidin derivatives; Cy, cyanidin derivatives; Pt, petunidin derivatives; Pg, pelargonidin derivatives; Pn, peonidin derivatives; Mv, malvidin derivatives; MonoOH, monohydroxylated anthocyanins; DiOH, dihydroxylated anthocyanins; TriOH, trihydroxylated anthocyanins; Met, methylated anthocyanins; Nonmet, nonmethylated anthocyanins; MonoGlc, monoglucoside anthocyanins; DiGlc, diglucosides anthocyanins; TriGlc, speculated triglucosides anthocyanins; Cf, caffeoyl anthocyanins; Cm, coumaryl anthocyanins; Ac, acetyl anthocyanins; Nonacy, nonacylated anthocyanins. The abbreviations of grape groups follow Figure 1.

**Figure 5 molecules-26-06696-f005:**
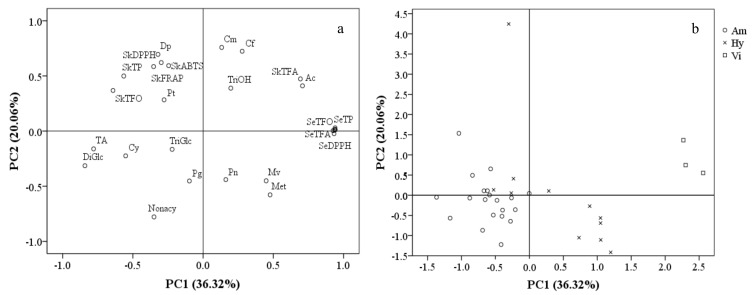
Distribution patterns of 27 variables (**a**) and 34 cultivars/accessions belonging to the three grape groups (**b**) 2017 and 2018 in the two−dimensional space of PC1 and PC2. The abbreviations of variable and grape groups follow Figure 1, Figure 2 and Figure 4.

**Table 1 molecules-26-06696-t001:** Grape cultivars/accessions collected in this study.

Pedigree	Number	Cultivars/Accessions
*Vitis amurensis*	20	Tonghua-1; Tonghua-3; Tonghua-7; Changbai-5; Changbai-8; Changbai-9; 92; 14; 75081; 73102; 84001; 84002; 85010; 086919; Zuoshan-1; Zuoshan-2; Shuangqing (Changbai-11); Shuangyou *; Shuanghong *; Shuangfeng *
Hybrids of *V. amurensis* and *V. vinifera*	11	Zuohong-1; Zuoyouhong; Beichun; Beihong; Beimei; Xuelanhong; Gongniang-1; Huapu-1; Beibinghong; Hasang; RS
*V. vinifera*	3	Cabernet Sauvignon; CabernetGernischt; Gem Cabernet

*: Intraspecific hybrids of *V. amurensis*.

**Table 2 molecules-26-06696-t002:** Retention times, MS/MS^2^
*m*/*z* values and mean contents (μg MGE/g DM) of anthocyanin compounds detected in the skins of grape cultivars/accessions belonging to the three study groups by Q-Exactive orbitrap MS.

Rt (min).	MS; MS2 (*m*/*z*)	Anthocyanin	*V. amurensis*		Interspecific Hybrids		*V. vinifera*	
			Mean	SE	Mean	SE	Mean	SE
2.41	789; 627,465,303	Dp-3,5,7-triglc *	35.96	21.63	6.37	5.53	—	—
2.97	773; 611,449,287	Cy-3,5,7-triglc *	26.88	15.48	4.19	3.31	—	—
3.01	627; 465,303	Dp-3,5-diglc	4608.23	3395.62	321.30	352.87	15.79	19.66
3.11	803; 641; 479,317	Pt-3,5,7-triglc *	26.15	18.45	15.14	32.68	—	—
3.48	611; 449,287	Cy-3,5-diglc	3229.93	2843.26	311.09	406.38	—	—
3.57	787; 625,463,301	Pn-3,5,7-triglc *	7.34	6.81	8.69	7.60	—	—
3.73	817; 655,493,331	Mv-3,5,7-triglc *	681.46	1404.39	33.16	20.69	—	—
3.72	641; 479,317	Pt-3,5-diglc	3157.22	2087.73	388.53	321.75	—	—
3.91	465; 303	Dp-3-glc	861.17	824.13	1261.65	1607.40	148.58	194.19
3.99	595; 433,271	Pg-3,5-diglc	16.01	10.77	3.38	2.32	—	—
4.18	625; 463,301	Pn-3,5-diglc	3027.85	2968.72	802.29	562.95	—	—
4.26	655; 493,331	Mv-3,5-diglc	17,344.52	6146.84	4273.00	2643.70	1.75	0.49
4.4	669; 507,465,303	Dp-3-acglc-5-glc	4.91	5.00	6.99	8.67	—	—
4.42	449; 287	Cy-3-glc	235.81	179.96	321.29	306.54	48.17	59.19
4.7	479; 317	Pt-3-glc	302.33	204.74	477.40	351.41	168.02	181.37
4.96	433; 271	Pg-3-glc	1.02	0.96	1.27	1.01	0.78	0.79
5.09	653; 491,449,287	Cy-3-acglc-5-glc	14.65	14.83	28.60	81.11	—	—
5.09	683; 521,479,317	Pt-3-acglc-5-glc	16.37	7.44	67.33	201.79	—	—
5.17	463; 301	Pn-3-glc	67.85	52.91	233.71	214.31	261.38	163.51
5.34	493; 331	Mv-3-glc	434.03	287.52	1350.04	867.34	2502.94	2426.91
5.6	667; 505,463,301	Pn-3-acglc-5-glc	15.93	17.60	6.84	3.98	—	—
5.6	697; 535,493,331	Mv-3-acglc-5-glc	71.59	38.15	32.77	27.89	—	—
5.61	963; 801,639,493,331	Mv-3-cmglc-5-glc-7-glc *	—	—	3.84	2.48	—	—
5.76	789; 627,465,303	Dp-3-cfglc-5-glc	—	—	17.65	39.97	—	—
5.73	507; 303	Dp-3-acglc	2.73	4.62	23.10	58.79	9.96	13.93
6.17	803; 641,479,317	Pt-3-cfglc-5-glc	—	—	9.02	15.20	—	—
6.18	773; 611,465,303 /773; 611,449,287	Dp-3-cmglc-5-glc /Cy-3-cfglc-5-glc ^#^	10.19	25.14	213.59	636.50	—	—
6.2	491; 287	Cy-3-acglc	1.01	0.94	3.72	3.18	2.96	4.21
6.33	521; 317	Pt-3-acglc	0.88	0.64	10.93	13.32	21.02	25.23
6.16	627; 303	Dp-3-cfglc	—	—	0.67	1.14	0.18	0.23
6.6	817; 655,493, 331	Mv-3-cfglc-5-glc	2.24	0.00	21.62	32.73	—	—
6.56	611; 287	Cy-3-cfglc	—	—	0.07	0.04	0.56	0.91
6.61	757; 595,449,287	Cy-3-cmglc-5-glc	1.07	1.49	23.54	35.97	—	—
6.62	787; 625,479,317 /787; 625,463, 301	Pt-3-cmglc-5-glc /Pn-3-cfglc-5-glc ^#^	4.24	8.92	54.50	107.03	—	—
6.84	505; 301	Pn-3-acglc	0.36	0.33	3.07	3.19	29.84	21.99
6.91	535; 331	Mv-3-acglc	1.83	1.68	22.46	29.21	191.51	38.03
6.73	641; 317	Pt-3-cfglc	—	—	0.25	0.29	0.46	0.60
6.93	611; 303	Dp-3-cmglc	0.51	0.56	2.25	4.85	8.49	14.50
7.07	771; 609,463,301	Pn-3-cmglc-5-glc	0.94	1.06	14.56	10.38	—	—
7.02	801; 639,493,331	Mv-3-cmglc-5-glc	1.10	1.08	94.32	142.02	—	—
7.07	625; 301	Pn-3-cfglc	—	—	0.19	0.23	4.53	6.12
7.23	655; 331	Mv-3-cfglc	—	—	0.57	1.50	18.88	15.31
7.37	595; 287	Cy-3-cmglc	—	—	0.13	0.24	3.62	6.06
7.48	625; 317	Pt-3-cmglc	—	—	0.69	1.18	14.31	23.56
8.25	609; 301	Pn-3-cmglc	—	—	0.36	0.69	29.73	42.93
8.37	639; 331	Mv-3-cmglc	0.00	0.00	4.60	10.86	197.95	256.04

* indicates the anthocyanin compound was speculated according to *m*/*z* values. And they were quantified using malvidin-3,5-*O*-diglucoside as standards. ^#^ indicates the two anthocyanin compounds were not separated according to *m*/*z* values.And they were quantified as the *p*-coumaroyl derivatives due to the fragment ion abundance. Abbreviations: Dp, delphinidin; Cy, cyanidin; Pt, petunidin; Pg, pelargonidin; Pn, peonidin; Mv, malvidin; triglc, triglucosides; diglc, diglucosides; glc, monoglucoside; acglc, (6-acetyl)-glucoside; cmglc, (6-coumaroyl)-glucoside; cfglc, (6-caffeoyl)-glucoside.

## Data Availability

The data presented in this study are available in Appendix A.

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
