# Peer review of "Quality Characteristics and Anthocyanin Profiles of Different Vitis amurensis Grape Cultivars and Hybrids from Chinese Germplasm"

_molecules, 2021, doi:10.3390/molecules26216696_

Round 1

Reviewer 1 Report

This manuscript presents an outstanding and well done study intending to assess the berry quality of the important and representative V. amurensis grapes and interspecific hybrids preserved in V. amurensis germplasm repository (VAGR) of Chinese Academy of Agricultural Sciences (CAAS) in Jilin province of China, and to compare with three red wine grapes of V. vinifera widely cultivated in the world.

For this purpose, the general characteristics, total phenolic contents and antioxidant activities as well as the anthocyanin contents and compositions were investigated.

The results of this research make a significant contribution in developing a comprehensive database related with quality and nutrition of berry and wine in the V. amurensis germplasm for breeding and commercialization.

This is a clear, concise, easy to understand and well-written manuscript. The research work is well designed and structured. The introduction is relevant and well documented. At the end of the Introduction, the purpose of this study is clearly shown. The methods are suitable and the obtained results are new and interesting being appropriately discussed and well highlighted and graphically presented. The conclusions have been appropriately formulated in a close relation with the obtained data. The number and quality of references are appropriate and relevant for the research topic.

Overall, this is a high quality manuscript, its content matches well with the journal’s purpose, but a minor revision is needed to improve the relevance and quality of the study.

Some recommendations in this purpose:

As regards the References, I kindly recommend to the authors to cite the references in the text according to the journal requirements (Please, see: https://www.mdpi.com/journal/molecules/instructions, where is mentioned that “In the text, reference numbers should be placed in square brackets [ ], and placed before the punctuation; for example [1], [1–3] or [1,3]”).

Also, in the section References, please use the Abbreviated Journal Name.

Please, replace “total phinolics” by “total phenolics” in the name of the Figures 2 and 5.

Author Response

Point 1: As regards the References, I kindly recommend to the authors to cite the references in the text according to the journal requirements (Please, see: https://www.mdpi.com/journal/molecules/instructions, where is mentioned that “In the text, reference numbers should be placed in square brackets [ ], and placed before the punctuation; for example [1], [1–3] or [1,3]”). Also, in the section References, please use the Abbreviated Journal Name.

Response 1: We have already revised the formats of the references in the text according to the journal requirements. And the reference numbers in the manuscript  have been marked with blue colour. But there were several references in Chinese without Abbreviated Journal Name.

Point 2: Please, replace “total phinolics” by “total phenolics” in the name of the Figures 2 and 5.

Response 2: We are very sorry for our spelling mistakes. We have already replaced “total phinolics” by “total phenolics” in the titles of Figures 2 and 5.

Reviewer 2 Report

Dear authors!

Dear Colleagues! 

The article is very interesting and comprehensive, but there are two small additions:

  1. In your research, in the Introduction, it is necessary to mention a large research about wild grape Vitis amurensis, collected in various geographical points in Russia [Razgonova et al., 2021]. An identified group of anthocyanin is also present in this research, so it is premature to talk about your superiority in the detection of anthocyanin in V. amurensis.
  2.  Is it possible to present Figure 5 in color, so it will be the most representative and informative.

Author Response

Point 1: In your research, in the Introduction, it is necessary to mention a large research about wild grape Vitis amurensis, collected in various geographical points in Russia [Razgonova et al., 2021]. An identified group of anthocyanin is also present in this research, so it is premature to talk about your superiority in the detection of anthocyanin in V. amurensis.

Response 1: Thanks for the suggestion of the reviewer. Razgonova et al. (2021) analyzed comprehensively the phenolic metabolites in Vitis amurensis grapes, including the wild grapes obtained from six different places in the Primorsky and Khabarovsk territories and the cultivated grapes obtained from the collection of N.I. Vavilov All-Russian Institute of Plant Genetic Resources. This study is very valuable for our research. We have added to present the important results of this study in line 26-28 of ‘1.Introduction’ [In a recent study, a total of 118 phenolic metabolites were detected in wild and cultivated V. amurensis grapes collected from Far East Russia (Razgonova et al. 2021).] and line 255-256 of ‘3.3.3 Composition of anthocyanins’[Razgonova et al. (2021) identified 7 acylated derivatives out of a total of 18 anthocyanin compounds in V. amurensis grapes from Far East Russia.], which were related to our research. And we have also added the citation of this study in line 225.

Point 2:  Is it possible to present Figure 5 in color, so it will be the most representative and informative.

Response 2: We are very sorry that we can’t really present Figure 5 in colour, which will increase the publishing charge and over the budget.

Reviewer 3 Report

The work is interesting and valuable, but it needs improvement and additions. Comments below:

- 2.1. Plant Material

is: „V. vinifra” should be „V. vinifera

- My guess is that for the V. vinifera varieties Cabernet Sauvignon; Cabernet Gernischt; Gem Cabernet a single sample was made which represented the V. vinifera species.

Why were these V. vinifera varieties chosen?

- Were the Determination of General Berry Characteristics (total soluble solid, titratable acidity and pH) determined in fresh or frozen material? In how many replicates were the analyses performed?

- In which acid was the titratable acidity expressed? It is useful to add this in the methodology.

- The description of sample preparation is unclear. First it is written that the fruits were kept at -20 degrees for further analyses (2.1. Plant Material). Then, the peels were separated from seeds by hand and freeze-dried (2.4. Pretreatment and Extraction of Phenolic Compounds) - this should be clarified.

- Why were peels and seeds used to prepare the extracts and not, for example, whole fruit?

- Was the distribution of results compatible with the normal distribution? What statistical test was used for this?

- What statistical test was used for correlation analyses?

- In the description of the results, the authors cite data for some varieties (page 5), while the data in Figure 1 represent average data (for all varieties).

- Are the values for individual grapes (Vitis amurensis, Iterspecific Hybrids, V. vinifera)  in Table S1 differ statistically significantly from each other?

- Table S1 shows values for 2017 or 2018?

- is: „Figure 2. Ranges and distributions of total phinolics” should be: „Figure 2. Ranges and distributions of total phenolics”

- Were the mean contents of anthocyanin compounds for the tested Vitis species presented in Table 2 statistically significantly different?

- In my opinion, the authors unnecessarily present values for two years:  2017 and 2018. If such data are presented, it would be useful to compare whether the values are statistically significantly different. Nevertheless, in the case of this article it does not make sense, because this was not the purpose of the study. For the purposes of this research, it would be useful to average the results for the individual data of these two years. The paper would have been clearer for the readers and at the same time it would not have lost its quality.

Author Response

Point 1: 2.1. Plant Material, “V. vinifra” should be “V. vinifera” .

Response 1: We are very sorry for our spelling mistakes. We have already replaced “V. vinifra” by “V. vinifera” in line 48 of “2.1. Plant Material”.

Point 2:  My guess is that for the V. vinifera varieties Cabernet Sauvignon; Cabernet Gernischt; Gem Cabernet a single sample was made which represented the V. vinifera species. Why were these V. vinifera varieties chosen?

Response 2: Because of the cold climate, the wine grapes of V. vinifera can’t be cultivated in the open field in Northeast China. We collected mature berries of three cultivars, ‘Cabernet Sauvignon’, ‘Cabernet Gernischt’ and ‘Gem Cabernet’, in a experimental vineyards of China Agricultural University in Beijing, managed by a cooperator. These cultivars with good adaptability and wine quality are widely cultivated and representative in the wine regions of Eastern China. And the three grapes of V. vinifera were analysed as the comparative group under the same experimental conditions. So we chose the three V. vinifera cultivars.

Point 3:  Were the Determination of General Berry Characteristics (total soluble solid, titratable acidity and pH) determined in fresh or frozen material? In how many replicates were the analyses performed?

Response 3: We are sorry for not detailing this part. The clusters picked from the grapevines were immediately placed with ice packs and taken to the adjacent laboratory of VAGR. In the lab, three 100-berry batches were randomly selected from the top, middle, and bottom portions of the clusters. Each batch was considered as one sample resulting in three replications for each grape cultivars/accessions. A half of fresh berries for each replicate sample were randomly selected to directly determine the general characteristics. The other half of berries were manually separated the skins and seeds. Then the frozen materials of grpe skins and seeds in refrigerated bags were bring back to the lab of Heilongjiang Bayi Agricultural University. The determinations of all the evaluation indicators were based on the three replications for each grape cultivars/accessions. So a total of 150 berries were measured and recorded the weights, transverse diameters and longitudinal diameters. And total soluble solid, pH and titratable acids, as well as contents and antioxidant activities of phenolic compounds (including anthocyanins) were determined with three replicates. According to the reviewer’s question, we have detailed the method in lines 55-56 of “2.1 Plant material” and lines 69-76 of “2.3 Determination of general berry characteristics”.

Point 4:  In which acid was the titratable acidity expressed? It is useful to add this in the methodology.

Response 4: We have added the detailed titration of titratable acids in lines 71-74 of “2.3 Determination of general berry characteristics”. It is accurate to express the titratable acids as a percentage. And we have revised in Fig. 1f and line 358 .

Point 5:  The description of sample preparation is unclear. First it is written that the fruits were kept at -20 degrees for further analyses (2.1. Plant Material). Then, the peels were separated from seeds by hand and freeze-dried (2.4. Pretreatment and Extraction of Phenolic Compounds) - this should be clarified.

Response 5: We are sorry that we don’t describe clearly the sample preparation. The detailed explanation is shown in the response to point 3. And we have revised “The fresh clusters were placed in refrigerated bags and taken to the laboratory.” into “The fresh clusters placed with ice packs were immediately taken to the laboratory.” in line 53 of “2.1 Plant material”. The fresh berries collected from the vines were placed with ice packs and immediately taken to the adjacent laboratory, the purpose was to keep fresh, not to freeze.

Point 6:  Why were peels and seeds used to prepare the extracts and not, for example, whole fruit?

Response 6: The primary aim of this study was to survey and evaluate the phenolic compounds, especially anthocyanins in V. amurensis grapes. The phenolic compounds exist maily in the grape skins and seeds. Moreover, the content and composition of anthocyanins in skins is a signature feature of V. amurensis. And the cultivars and hybrids of V. amurensis are mainly used to made wines. The wine phenolics from grape skins are very important for the quality and nutrition of the wines. The seed phenolics are useful to health care products and additives. So it was more targeted to extract and analyse the phenolics with separated skins and seeds. And general characteristics (total soluble solid, pH and titratable acids) were used to assess the grape juice in this study.

Point 7:  Was the distribution of results compatible with the normal distribution? What statistical test was used for this?

Response 7: Yes, all the data of the variances were compatible with the normal distribution using with Kolmogorov-Smirrow and Q-Q figure.

Point 8:  What statistical test was used for correlation analyses?

Response 8: The correlations of different indicators were analyzed with Pearson at the 95% confidence level. And we have also added the statistical test in lines 120-127 of “2.8 Statistical Analysis”.

Point 9:  In the description of the results, the authors cite data for some varieties (page 5), while the data in Figure 1 represent average data (for all varieties).

Response 9: The salient data of several grape cultivars/accessions were mentioned in the description of the results. But we didn’t list the data of each cultivars/accessions in tables and figures, which might make the manuscript or supplementary file too long, because there were too many grape cultivars/accessions (34) and indicators (12 except for anthocyanins). So the grapes were divided into three groups, including Vitis amurensis, V. vinifera and interspecific hybrids. The ranges and distributions of general berry characteristics were shown in Figure 1. The ranges and distributions of total phenolic contents and antioxidant activities were shown in Figure 2. And the average data of two years were shown in Table S1.

Point 10:  Are the values for individual grapes (Vitis amurensis, Interspecific Hybrids, V. vinifera)  in Table S1 differ statistically significantly from each other?

Response 10: Thanks very much for the reviewer’s suggestion. Because all the data of the variances were compatible with the normal distribution, but most were not homogeneous. The data of the same indicator among different grape groups were subjected to Welch's ANOVA with Games-Howell at the 95% confidence level. And we added the results in Table S1.

Point 11:  Table S1 shows values for 2017 or 2018?

Response 11: The data were the average data of the two years in Table S1. And we have revised the title of Table S1.

Point 12:  is: “Figure 2. Ranges and distributions of total phinolics” should be: “Figure 2. Ranges and distributions of total phenolics”

Response 12: We are very sorry for our spelling mistakes. We have already replaced “total phinolics” by “total phenolics” in the titles of Figures 2 and 5.

Point 13:  Were the mean contents of anthocyanin compounds for the tested Vitis species presented in Table 2 statistically significantly different?

Response 13: Thanks very much for the reviewer’s suggestion. But we think it is more meaningful to analyze the anthocyanin composition with ANOVA. Because there were too many anthocyanin compounds detected so that it was hard to find the difference rule among the grape groups. We have added the ANOVA results of anthocyanin composition in Figure 4. The ANOVA method was the same as Table S1.

Point 14:  In my opinion, the authors unnecessarily present values for two years:  2017 and 2018. If such data are presented, it would be useful to compare whether the values are statistically significantly different. Nevertheless, in the case of this article it does not make sense, because this was not the purpose of the study. For the purposes of this research, it would be useful to average the results for the individual data of these two years. The paper would have been clearer for the readers and at the same time it would not have lost its quality.

Response 14: Thanks very much for the reviewer’s suggestion.  The suggestion is valuable and helpful to improve our manuscript.  And we have revised into the average data of the two years in Figure 1, 2, 4 and 5. We also made the corresponding revision in the manuscript.

Reviewer 4 Report

Title: Berry Quality Characteristics of Vitis amurensis Germplasm in China

This study compares physical attributes (weight and sizes), chemical (pH, soluble solid contents, acidity, phenolic, and anthocyanin profiles), and antioxidant properties of grapes from different species. Grapes from 31 different Vitis amurensis cultivars, grown mainly in Asia, 3 traditional Vitis vinifera wine grapes, and 11 hybrids from crossing between V. amurensis and V. vinifera were accessed on this study.

Although the study provides extensive data on the quality attributes of different grape cultivars, it lacks organization and isn’t easy to read. Adding line numbers would be helpful for the reviewer. English also needs to be polished, and many sentences are difficult to understand.

The title of the manuscript should be reconsidered. A suggestion: Quality characteristics and anthocyanin profiles of different Vitis amurensis grape cultivars and hybrids from Chinese germplasm

The introduction is too long, and the importance of the study is not clearly explained. The results and graphic presentations are not clear. Results should be separated from Discussion, that is, two different sections.

The authors talk about rain affecting the quality of grapes but offer no further explanation.

What is the commercial importance of Vitis amurensis and the significant differences between the common wine grape cultivars?

Titratable acidity should be presented as a percentage.

Correct “phinolics” to “phenolics”.

Color of the different grape berries? Was it different? If the anthocyanin content of the skin was different between cultivars, I would expect the color of the grapes to be different. Was it measured?

Figure 4 should not be split.

Lacks discussion of significant findings.

Author Response

Point 1: Although the study provides extensive data on the quality attributes of different grape cultivars, it lacks organization and isn’t easy to read. Adding line numbers would be helpful for the reviewer. English also needs to be polished, and many sentences are difficult to understand.

Response 1: Thanks very much for the reviewer’s suggestion. We have added the line numbers and page numbers.  And we have also revised and polished the whole manuscript marked with red color.

Point 2: The title of the manuscript should be reconsidered. A suggestion: Quality characteristics and anthocyanin profiles of different Vitis amurensis grape cultivars and hybrids from Chinese germplasm.

Response 2: Thanks very much for the reviewer’s suggestion. The suggested title is more accurate and comprehensive than the original title, and highlights the anthocyanin profiles. We have revised the manuscript title into “Quality characteristics and anthocyanin profiles of different Vitis amurensis grape cultivars and hybrids from Chinese germplasm”.

Point 3: The introduction is too long, and the importance of the study is not clearly explained. The results and graphic presentations are not clear. Results should be separated from Discussion, that is, two different sections.

Response 3: Thanks very much for the reviewer’s suggestion. We have revised the introduction in lines 12-24 and re-written the importance of the study in lines 23-24 and 39-44. In order to be clearer for the readers, we have revised into the average data of the two years in Figure 1, 2, 4 and 5. We also made the corresponding revision in the lengends of the figures. We have separated Discussion in lines 325-412 from Results, and separated “3.5 Correlation analysis” in lines 281-301.

Point 4: The authors talk about rain affecting the quality of grapes but offer no further explanation.

Response 4: Thanks very much for the reviewer’s suggestion. The purpose of this research was to investigate the quality characteristics and anthocyanin profiles of different Vitis amurensis grapes, but not the effects of climate on the berry quality. In order to be clearer for the readers, we have revised into the average data of the two years in Figure 1, 2, 4 and 5. We also deleted the difference between the two years in the muanuscript.

Point 5: What is the commercial importance of Vitis amurensis and the significant differences between the common wine grape cultivars?

Response 5: The cold resistance is the most prominent advantage of Vitis amurensis, whose roots can survive at extremely low temperatures of -45°C. In winter, the grapes with Vitis amurensis pedigree needn’t to be cover with soil in most areas of the North China. While the V. vinifera grapes can’t be cultivated in the open field of Heilongjiang, Jilin and northern Liaoning. So V. amurensis is one of the precious rootstock and breeding resources with cold resistance in Vitis.

The V. amurensis cultivars and hybrids are suitable to cultivated in the cold regions. The cost of their cultivation is relatively low. They are cultivated in the North of China, including Heilongjiang, Jilin, Liaoning and Hebei provinces, as well as Beijing. Most of them are used to make wines, including dry wines, sweet wines, ice wines and low-alcohol and non-alcoholic wines. The wine pomace can be produced to health care products and additives. And several interspecific hybrids are table grapes or juice grapes. The usage of V. amurensis grapes have been added in lines 15-18 and 19-23 of Introduction.

Point 6: Titratable acidity should be presented as a percentage.

Response 6: Thanks very much for the reviewer’s suggestion. It is accurate to express the titratable acids as a percentage. And we have revised in Fig. 1f and line 358.

Point 7: Correct “phinolics” to “phenolics”.

Response 7: We are very sorry for our spelling mistakes. We have already replaced “total phinolics” by “total phenolics” in the titles of Figures 2 and 5.

Point 8: Color of the different grape berries? Was it different? If the anthocyanin content of the skin was different between cultivars, I would expect the color of the grapes to be different. Was it measured?

Response 8: Thanks very much for the reviewer’s suggestion. The colour of the grapes should be measured, and analysed the correlation with anthocyanin content and composition. But because of negligence, we didn’t measure the colour of the grapes. And there aren’t fresh materials reserved to supplement the experiment. We are so sorry about that.

Point 9: Figure 4 should not be split.

Response 9: We don’t understand the suggestion well enough. We have revised into the average data of the two years and added the results of ANOVA in Figure 4.

Point 10: Lacks discussion of significant findings.

Response 10: Thanks very much for the reviewer’s suggestion. We have added the discussion of significant findings in lines 344-347 and 372-412.

Reviewer 5 Report

While the general information on V. amurensis antioxidant properties and polyphenolics content was reported previously, this study aimed to synoptically describe the characteristics of individual cultivars. Nevertheless, the data shown are grouped for V. amurensis, V. vinifera and their hybrids. Thus, there is no fundamental contribution of this manuscript to the basic knowledge.

Minor comments:

  1. Please check and improve the English throughout the text.
  2. Are all studied cultivars of V. amurensis red cultivars? It is well known that there is a big difference between red and white grapes form the point of view of their polyphenolic content and antioxidant properties. Grouping both types of cultivars together would distort the results.
  3. Where is the difference between skin and seed characteristics shown in the figure 1?
  4. Please improve the individual panel headings in the figure 2.

Author Response

Point 1: Please check and improve the English throughout the text.

Response 1: Thanks very much for the reviewer’s suggestion. We have checked and revised the English throughout the manuscript.

Point 2: Are all studied cultivars of V. amurensis red cultivars? It is well known that there is a big difference between red and white grapes form the point of view of their polyphenolic content and antioxidant properties. Grouping both types of cultivars together would distort the results.

Response 2: In this study, all the collected materials were red grapes. Because all the wild cultivars/accessions and the core germplasm of V. amurensis are red grapes. Only a few interspecific hybrids are white grapes. Moreover, the outstanding feature of V. amurensis grapes was the abundant anthocyanin contents, which was the major aim of this study.

Point 3: Where is the difference between skin and seed characteristics shown in the figure 1?

Response 3: We are very sorry for our mistakes. The data of skins and seeds were not shown in Figure 1. We have deleted “Sk, grape skin; Se, grape seed”.

Point 4: Please improve the individual panel headings in the figure 2.

Response 4: We are very sorry for our spelling mistakes. We have already replaced “total phinolics” by “total phenolics” in the titles of Figures 2. And we have added the explanation of the abbreviations.

Round 2

Reviewer 4 Report

After revision, the manuscript was significantly improved.

This manuscript is a resubmission of an earlier submission. The following is a list of the peer review reports and author responses from that submission.